# Enhancing the Productivity and Stability of Superoxide Dismutase from *Saccharomyces cerevisiae* TBRC657 and Its Application as a Free Radical Scavenger

**Phitsanu Pinmanee** [1,2], **Kamonwan Sompinit** [2], **Jantima Arnthong** [2], **Surisa Suwannarangsee** [2], **Angkana Jantimaporn** [3], **Mattaka Khongkow** [3], **Thidarat Nimchua** [2] and **Prakit Sukyai** [1,*]

1   Biotechnology of Biopolymers and Bioactive Compounds Special Research Unit, Department of Biotechnology, Faculty of Agro-Industry, Kasetsart University, Bangkok 10900, Thailand; phitsanu.pin@biotec.or.th
2   Enzyme Technology Research Team, National Center of Genetic Engineering and Biotechnology (BIOTEC), Pathum Thani 12120, Thailand; kamonwan.sompinit@gmail.com (K.S.); jantima.arn@biotec.or.th (J.A.); surisa.suw@biotec.or.th (S.S.); thidarat.nim@biotec.or.th (T.N.)
3   Nanomedicine and Veterinary Research Team, National Center of Nanotechnology (NANOTEC), Pathum Thani 12120, Thailand; angkana.jan@ncr.nstda.or.th (A.J.); mattaka@nanotec.or.th (M.K.)
*   Correspondence: fagipks@ku.ac.th

**Abstract:** Superoxide dismutase (SOD) is crucial antioxidant enzyme that plays a role in protecting cells against harmful reactive oxygen species (ROS) which are generated inside cells. Due to its functionality, SOD is used in many applications. In this study, *Saccharomyces cerevisiae* TBRC657 was selected as the SOD producer due to its high SOD production. After investigating an optimized medium, the major components were found to be molasses and yeast extract, which improved SOD production up to 3.97-fold compared to a synthetic medium. In addition, the optimized medium did not require any induction, which makes it suitable for applications in large-scale production. The SOD formulation was found to increase the stability of the conformational structure and prolong shelf-life. The results show that 1.0% (*w/w*) trehalose was the best additive, in giving the highest melting temperature by the DSF method and maintaining its activity at more than 80% after storage for 6 months. The obtained SOD was investigated for its cytotoxicity and ROS elimination against fibroblast cells. The results indicate that the SOD enhanced the proliferation and controlled ROS level inside the cells. Thus, the SOD obtained from *S. cerevisiae* TBRC657 cultured in the optimized medium could be a candidate for use as a ROS scavenger, which can be applied in many industries.

**Keywords:** superoxide dismutase; *Saccharomyces cerevisiae*; molasses; enzyme production; enzyme formulation; ROS elimination

## 1. Introduction

Superoxide dismutase (SOD) (EC 1.15.1.1) is one of the essential antioxidant enzymes that play a role in protecting cells against harmful reactive oxygen species (ROS), especially superoxide radicals ($O_2{}^-$), which are generated during stress conditions [1,2]. SOD catalyzes the dismutation of superoxide radicals to hydrogen peroxide ($H_2O_2$) and molecular water ($H_2O$) whilst hydrogen peroxide is continually converted to water by catalase (CAT) or glutathione peroxidase (GPx), which already exist inside all organisms. The combination of these enzymes controls the levels of the harmful free radicals [3]. In addition, SOD eliminates not only the endogenous free radicals which are produced during oxidative metabolism, but also eliminates the exogenous oxidants which are generated by exposure to stimuli such as UV light or some chemicals [4]. SOD is a metalloenzyme which requires a metal ion as a cofactor for its activity. SOD is classified by the metal ion that binds at the active site of the enzyme: Cu/Zn-SOD, Mn-SOD, Fe-SOD and Ni-SOD [5]. In general, SODs are found in all organisms exposed to oxygen including microorganisms, animals

and plants, and can be extracted and utilized for ROS elimination [6]. Based on the antioxidant activities of SOD as a free radical scavenger, in oxidative stress protection and anti-inflammatory activity, SOD is widely used in many applications such as in the medical, pharmaceutical, agricultural, food and cosmetic industries, resulting in increasing enzyme use every year [7–10].

Due to this reason, there are many studies related to SOD production obtained from microorganisms, including wild type strains such as *S. cerevisiae*, *Kluyveromyces marxianus*, *Phanerochaete chrysosporium* and *Aspergillus glaucus*, including recombinant strains. Moreover, in those studies the target enzyme was normally produced by using a laboratory medium, which might not be suitable for large-scale production [11–16]. However, for industrial applications, the economic reasons behind the enzyme production process from selected microorganisms is crucial, which should be the first priority in developing commercial microbial-based products. The major goal is to discover a low-cost medium that can produce a high amount of microbial cell biomass without any effect on the biological functions. Moreover, the optimized medium must provide the essential nutrients to microorganisms which are necessary for their metabolism, growth and metabolite production. The trend for using low-cost products and/or by-products such as molasses, potato peels, or food waste as medium components has increased due to the high number of bioactive compounds which can act as carbon-sources and nitrogen-sources and could be utilized by microorganisms via biological processes for their growth and enzyme production. Thus, the optimization of the production medium has been widely investigated by using those by-products with a possibility to apply the production process to large-scale production [17–19]. Otherwise, SOD is a labile enzyme which easily degrades when exposed to the environment [20]. Thus, enzyme formulation technology has been used with the aim of adding the target enzyme with the appropriate concentration and types of additives to stabilize its conformational structure and maintain its biological function during storage conditions [21]. Enzyme formulation is a very important step in addition to enzyme production and the downstream processes which play a role in the final usage and success of the enzyme product. Ideally, enzymes should not need formulation, which is an additional step in enzyme production process resulting in a higher cost of the final product. However, almost all enzyme products are unstable due to their properties, structures and weaknesses in one or more ways. Thus, the usefulness of enzymes in industry will increase and facilitate with some better properties such as a high activity, function at broad range of pH and temperature values, heat resistance, low production cost and long storage life [22–24]. For this reason, mixing the target enzyme with some additives may alter the conformational structure of the target enzyme, resulting in a more compact structure and more resistance to the environment. In contrast, some additives might have a strong effect on the structure of enzymes, leading to a loss of biological function. Thus, the concentration and type of additives must be evaluated in order to obtain suitable additives for protecting the structure of enzymes during operating conditions and prolonging its storage stability [25–27].

In this study, the optimization of SOD production from *S. cerevisiae* TBRC657, which possesses high levels of SOD, was studied in order to assess the optimized medium and production process, with the potential to be applied to large-scale production. After that, a SOD formulation with various additives was examined to discover the suitable concentration and types of additives which could stabilize the SOD structure when exposed to harsh environments by differential scanning fluorometry (DSF). Then, the activities of selected formulated SODs were measured after having been stored at 4 °C for 6 months in order to investigate a potential formulated enzyme for long-term storage which can be applied in a final commercial product. In addition, the cytotoxicity and antioxidant properties of the obtained SOD were tested against fibroblast cells to confirm its function on ROS elimination in hydrogen peroxide-treated cells. The results suggest that the obtained SOD from *S. cerevisiae* TBRC657 can be used as an antioxidant ingredient which can be used in many industrial applications.

## 2. Materials and Methods

### 2.1. Microbial Isolates and Media Preparation

Approximately 150 isolates of baker's yeast (*Saccharomyces* sp.) which were isolated from various habitats in Thailand were used in this study. Those isolates were deposited and can be accessed at the Thailand Bioresources Research Center (TBRC), Pathum Thani, Thailand. The culture media used in this study were (1) a YPD medium consisting of 1.0% (*w/v*) yeast extract, 2.0% (*w/v*) peptone and 2.0% (*w/v*) glucose and (2) an optimal medium consisting of 3.0% (*w/v*) yeast extract and 25.0% (*w/v*) molasses.

### 2.2. Screening for SOD Producer

A single colony of each isolate was inoculated in 5 mL of YPD in a 50 mL test tube for use as the starting culture. The starters were cultivated at 30 °C with a shaking speed of 250 rpm for 16 h. Then, 0.5 mL of fresh starter was inoculated in 50 mL of YPD in a 250 mL Erlenmeyer flask. All cultures were cultivated at 30 °C with a shaking speed of 250 rpm for 24 h. After that, the cells were harvested by centrifugation at $5000\times g$ for 10 min and washed twice with a PBS buffer. Then, the cells were resuspended with 50 mM sodium phosphate buffer (pH 7.5) with a ratio of wet cell weight to buffer volume of 1 g to 4 mL. The cell suspension was disrupted by glass beads with BeadBug™ microtube homogenizer (Benchmark Scientific, Sayreville, NJ, USA) for 5 passages using a speed of 4000 rpm for 1 min. The supernatant was harvested by centrifugation at $25,000\times g$, 4 °C for 1 h. The crude extract was measured for SOD activity (U/mL) at pH 7.0 and 37 °C with a SOD determination kit (Sigma-Aldrich, St. Louis, MO, USA). One unit (U) of enzyme was defined as the amount of required to inhibit the optical density of 440 nm of WST-1 formazan formation under assay conditions. The protein concentration (mg/mL) was determined by Bradford solution (Bio-Rad, Hercules, CA, USA) using bovine serum albumin (BSA) as the standard.

### 2.3. Optimization of Temperature for SOD Production

The effect of culturing temperature was investigated to determine the optimal SOD production of *S. cerevisiae* TBRC657. To obtain the optimal temperature, *S. cerevisiae* TBRC657 was cultured in 50 mL YPD medium in 250 mL Erlenmeyer flasks. Then, the cells were cultivated at different temperatures ranging from 25–40 °C with a shaking speed of 250 rpm for 24 h. The crude extract from each experiment was collected and measured for SOD activity and protein concentration as described before. For each treatment, three independent replicates were performed.

### 2.4. Effect of Type and Concentration of Carbon-Source for SOD Production

A culture starter of *S. cerevisiae* TBRC657 was cultured as described before. Then, the optimization of the carbon source was studied by culturing the starter in 250 mL Erlenmeyer flasks with 50 mL production medium which consisted of 1.0% (*w/v*) yeast extract and 2.0% (*w/v*) peptone and was supplemented with 2.0% (*w/v*) of various carbon sources—glycerol, sucrose, fructose, maltose, lactose, raffinose, corn starch, molasses or sugarcane bagasse hydrolysate (kindly provided by a sugarcane mill in Thailand). The cells were cultivated at 35 °C with a shaking speed of 250 rpm for 24 h. The production medium containing 2.0% (*w/v*) glucose (normal YPD) was used as a control. To examine the optimal concentration of selected carbon sources, the production medium consisted of 1.0% (*w/v*) yeast extract and 2.0% (*w/v*) peptone and was supplemented with 5.0% (*w/v*)–30.0% (*w/v*) molasses. The cells were cultivated at 30 °C with a shaking speed of 250 rpm for 24 h. The cell harvesting and SOD extraction were carried out as described before. For each tested medium, three independent replicates were analyzed.

### 2.5. Effect of Type of Nitrogen-Source for SOD Production

A culture starter of *S. cerevisiae* TBRC657 was cultured as described before. Then, the optimization of the nitrogen source was studied by culturing the starter in 250-mL

Erlenmeyer flasks with 50 mL production medium which consisted of 2.0% (*w/v*) glucose and was supplemented with 3.0% (*w/v*) of various nitrogen sources, which were peptone, tryptone, yeast extract, malt extract, soytone, urea, sodium nitrate, ammonium sulfate or albumin. The cells were cultivated at 35 °C with a shaking speed of 250 rpm for 24 h. The normal YPD production medium was used as a control. The cell harvesting and SOD extraction were carried out as described before. For each experiment, three independent replicates were analyzed.

### 2.6. Effect of Type and Concentration of Inducer for SOD Production

A culture starter of *S. cerevisiae* TBRC657 was cultured as described before. Then, the optimization of inducers—hydrogen peroxide and menadione—was studied by culturing the starter in in 250-mL Erlenmeyer flask with 50 mL optimal production medium, which consisted of 25.0% (*w/v*) molasses and 3.0% (*w/v*) yeast extract. The cells were cultivated at 35 °C with a shaking speed of 250 rpm. After 16 h of cultivation, 100 mM hydrogen peroxide or 1 mM menadione were added to the culture. Then, all experiments were continually cultivated at the same conditions for 24 h. The optimal production medium without chemical induction was used as a control. The cell harvesting and SOD extraction were carried out as described before. For each experiment, three independent replicates were analyzed.

### 2.7. Optimization of SOD Production in 5 L Bioreactor

To scale-up the SOD production obtained from *S. cerevisiae* TBRC657 at a laboratory scale, enzyme production in a BIOSTAT® B Plus 5 L- bioreactor (Sartorius, Germany) was investigated. The starter was prepared by inoculating a single colony in 20 mL YPD medium in 50-mL Erlenmeyer flasks, and inoculated at 35 °C, 250 rpm for 16 h. Then, the obtained cultivation was transferred in 180 mL YPD medium to a 1 L Erlenmeyer flask, and cultured at 30 °C, 250 rpm for 16 h, which was further used as starter in the 5 L fermenter. After that, the obtained starter was transferred to 1.8 L of production medium consisting of 25.0% (*w/v*) molasses and 3% (*w/v*) yeast extract which had already been prepared in the 5 L bioreactor. The cells were cultured at 35 °C for 24 h with an agitation rate of 300 rpm and different aeration rates ranging from 0.0–1.0 vvm. Cell harvesting and SOD extraction were carried out as described before.

### 2.8. SOD Formulation by Thermal Shift Assay and Storage Stability

Enzyme formulation was investigated for enhancing its storage stability by mixing the obtained SOD with suitable chemicals at different concentrations in order to stabilize conformational structure and prolong shelf-life of a formulated SOD. To investigate the effect of additives on SOD stability, the change of the conformational structure of the enzyme was investigated by differential scanning fluorescence (DSF). The reaction of the DSF assay consisted of (1) 2 μL of 1 mg/mL partial purified SOD, (2) 2 μL SYPRO$^{TM}$ Orange dye (S5692, Sigma-Aldrich, St. Louis, MO, USA), (3) 5–15 μL of test additives (Table 1) at different concentrations ranging from 0.1% (*w/v*)–10% (*w/v*) and (4) sterile Milli-Q® water up to a reaction volume of 20 μL. Then, all reactions were incubated in CFX Real-Time PCR (Bio-Rad, Hercules, CA, USA) and the equipment system was set to a Förster Resonance Energy Transfer (FRET) scanning mode by gradually increasing the temperature by 0.5 °C per 30 s. The fluorescence signal was measured for excitation wavelength at 490 nm and emission wavelength at 570 nm. Then, the melting temperature ($T_m$) obtained from each formulate was monitored and the difference of the melting temperature ($\Delta T_m$) was estimated by comparing with a $T_m$ obtained from an unformulated control. All samples were carried out in triplicate.

**Table 1.** List of chemicals used in the thermal shift assay.

| Group | Chemicals |
| --- | --- |
| Polyols and Polymers | Ethylene glycol |
| | Polyethylene glycol 1000 (PEG 1000) |
| | PEG 3350 |
| | PEG 8000 |
| | Glycerol |
| Salts | Sodium chloride |
| | Potassium chloride |
| | Calcium chloride |
| | Magnesium chloride |
| | Ammonium sulfate |
| | Manganese (II) chloride |
| Sugar alcohol | Mannitol |
| | Sorbitol |
| | Xylitol |
| Sugar | Fructose |
| | Maltose |
| | Sucrose |
| | Trehalose |
| | Mannose |

Following this, the formulates which exhibited the top 5 highest $\Delta T_m$ were analyzed for their biological function by measuring their activities compared to unformulated SOD as described before. Then, all selected formulates were investigated for their storage stabilities by storing all formulates at 4 °C. The remaining activity obtained from each formulate was measured after having been kept at 4 °C for a period of 1–6 months compared with SOD activity obtained from an unformulated control at day 0 which was set as 100%. All samples in this experiment were carried out in triplicate.

### 2.9. Cytotoxic Activity of SOD on Fibroblasts

To determine the toxicity of the SOD, a cytotoxicity assay was performed against fibroblasts cell. Approximately 18,000 fibroblast cells were cultured in Dulbecco's Modified Eagle Medium (DMEM) media supplemented with 4.5 mg/mL glucose and 10% (*v/v*) Fetal Bovine Serum (FBS) in 96-well plates. Then, the fibroblast cells were maintained in an optimum condition at 37 °C in a humidified 5% $CO_2$ incubator for 24 h. Following this, the fibroblast cells were treated with various dosages of crude SOD ranging from 50–500 units, whilst the untreated cells were used as a control. All reactions were continually incubated at 37 °C in a humidified 5% $CO_2$ incubator for 24 h. After this, the cell viability was evaluated with an MTT assay (Sigma-Aldrich, St. Louis, MO, USA). The cell viability (%) against the untreated cells was calculated. For each dosage, three independent replicates were performed.

### 2.10. Protective Effect of SOD on Oxidative Stress-Induced Fibroblasts

The protective effect of crude SOD on ROS was examined by adding exogenous hydrogen peroxide to fibroblast cells. Fibroblast cells were cultured and treated with various dosages of SOD, as mentioned before. After the SOD treatment, various concentrations of hydrogen peroxide ranging from 0–200 μg/mL were individually added to the SOD-treated fibroblast cells, then continually incubated at 37 °C in a humidified 5% $CO_2$ incubator for 1 h. Hydrogen peroxide was removed from the reactions and the fibroblast cells were washed with a PBS buffer twice. Cell viability was measured by MTT assay as described before. The efficiency of SOD–treated fibroblast cells was plotted between the cell viability obtained from different SOD dosages and different concentrations of hydrogen peroxide representing the protective effects of SOD on treated fibroblasts compared to the untreated fibroblast cells as a control.

*2.11. Statistical Analysis*

All the data given in the study are means of the three duplicates $\pm$ standard deviation. One-way ANOVA followed by Tukey's Multiple Range Tests were performed using SPSS 11.5 at the 5% significance level.

## 3. Results and Discussion

*3.1. Selection of SOD Producer*

In this study, to obtain the microorganisms producing the highest SOD level, all strains were screened for their SOD productivity in YPD medium under the same conditions. As expected, all strains of baker's yeast which were selected for this study were able to produce SOD inside their cells at different enzyme production levels. Among those, *S. cerevisiae* TBRC657 exhibited the highest activity of SOD at 207.14 U/mL. In this study, the different strains of yeast cells exhibited different levels of SOD production and purities. Among those, *S. cerevisiae* TBRC657 showed the highest SOD production but also produced a high concentration of the other proteins, which can be calculated as not being the highest value of specific SOD activity when compared to others (data not shown). A higher specific enzyme activity indicates a higher purity of the target enzyme, which leads to easier downstream processes, especially in the purification step [28]. Thus, this strain, categorized as the GRAS strain, with a potential for many applications was selected for further research.

*3.2. Optimization of Temperature for SOD Production*

Culturing temperature is the first factor for optimizing a cell biomass and SOD production, which represents the development and functionality of those microorganisms [29,30]. Thus, the results of cell growth and SOD production in YPD medium obtained from culturing at various temperatures were different, as shown in Table 2. After 24 h of cultivation, the highest cell growth and SOD production of *S. cerevisiae* TBRC657 was obtained at 35 °C, at values of 1.23 g and 220.98 U/mL, respectively. The temperatures giving the second and third-highest SOD production were 30 °C and 25 °C (205.14 and 146.16 U/mL, respectively). Meanwhile, only a small amount of cell growth and SOD production was detected when culturing at 45 °C. According to the obtained results, 35 °C was the suitable temperature to be used in further experiments. In addition, this temperature was deemed suitable for enzyme production via manufacturers in Thailand due to the average yearly temperature which range from 33–38 °C (data provided by a private company).

**Table 2.** Effect of temperature on SOD production from *S. cerevisiae* TBRC657.

| Temperature | Total Cell Weight | SOD Activity |
|---|---|---|
| 25 | 1.12 $\pm$ 0.08 | 146.16 $\pm$ 10.87 [b] |
| 30 | 1.22 $\pm$ 0.05 | 205.14 $\pm$ 9.75 [a] |
| 35 | 1.23 $\pm$ 0.04 | 220.98 $\pm$ 8.41 [a] |
| 40 | 0.57 $\pm$ 0.01 | 1.57 $\pm$ 0.03 [c] |

Different superscripts within the same column indicate significant differences ($p < 0.05$) and values are presented as mean $\pm$ SD ($n = 3$).

*3.3. Optimization of Medium Composition for SOD Production at Flask-Scale*

To improve the cell growth and SOD production of *S. cerevisiae* TBRC657, an optimization of the fermentation process involving carbon sources, nitrogen sources or inducer types and concentrations were individually investigated at flask-scale. After culturing the selected strain in different carbon-source base media, the cell weight and SOD activity were also different (Table 3). When molasses was applied as the major carbon-source, the growth of *S. cerevisiae* TBRC657 and its SOD production were the highest (1.86 g and 328.24 U/mL, respectively). The second and third most effective carbon-sources for a higher SOD activity were glucose (normal YPD) and sucrose, respectively. In contrast, lower values of SOD production were obtained when culturing *S. cerevisiae* TBRC657 with corn starch (18.19 U/mL). According to the results, molasses showed the highest potential for use as a

low-cost carbon-source for the enzyme production, which increased the SOD activity up to 1.58-fold when compared to the normal YPD (328.24 and 208.18 U/mL, respectively). In addition, molasses obtained from sugarcane mills represents a low-cost source of waste containing a high concentration of mono- and disaccharides such as glucose, sucrose and fructose (approximately 50%(*w/w*)), which microorganisms can utilize for their growth and producing the essential metabolites [31,32]. As for the other carbon-sources, fructose and sucrose were also good sources for cell biomass and SOD production due to the ability of the yeast cell to utilize those carbon-sources. On the other hand, *S. cerevisiae* TBRC657 could not utilize starch for its growth, which might require additional protocols such as applying starch-degrading enzymes to the system [33]. Those additional protocols may affect to the overall cost of enzyme production and are not easy to operate at the pilot scale. Thus, the medium consisting of molasses as a major carbon-source gave the greatest values of cell weight and SOD activity and was used in further experiments.

**Table 3.** Effect of carbon source on SOD production from *S. cerevisiae* TBRC657.

| Carbon Source | Total Cell Weight | SOD Activity |
|---|---|---|
| Glucose (YPD) | 1.23 ± 0.03 | 208.18 ± 10.87 [b] |
| Glycerol | 0.95 ± 0.05 | 76.71 ± 9.75 [d] |
| Sucrose | 1.19 ± 0.04 | 206.86 ± 12.4 [b] |
| Fructose | 1.23 ± 0.03 | 190.30 ± 10.42 [b,c] |
| Maltose | 0.99 ± 0.02 | 90.63 ± 5.18 [d] |
| Lactose | 0.33 ± 0.01 | 77.25 ± 2.15 [d] |
| Raffinose | 0.89 ± 0.02 | 98.65 ± 5.53 [d] |
| Corn starch | 0.65 ± 0.02 | 18.19 ± 2.71 [e] |
| Molasses | 1.86 ± 0.03 | 328.24 ± 29.44 [a] |
| Bagasse hydrolysate | 0.42 ± 0.01 | 23.78 ± 1.03 [e] |

Different superscripts within the same column indicate significant differences ($p < 0.05$) and values are presented as mean ± SD ($n = 3$).

To examine the optimum concentration of molasses as a major carbon-source, *S. cerevisiae* TBRC657 was cultured at different concentrations of molasses ranging from 5.0% (*w/w*)–30.0% (*w/w*). The result showed that the highest cell growth was obtained when culturing *S. cerevisiae* TBRC657 with 15.0%(*w/w*) molasses, which provided a SOD production of 350.88 U/mL (Table 4). On the other hand, a molasses concentration of 25.0% (*w/w*) provided the highest SOD production of 590.87 U/mL. The second and third highest SOD productions were achieved with 30 (*w/w*) and 20.0% (*w/w*) molasses, respectively. Thus, 25.0% (*w/w*) molasses was selected as the optimum concentration for use as a sole carbon-source for SOD production from *S. cerevisiae* TBRC657.

**Table 4.** Effect of molasses concentration on SOD production from *S. cerevisiae* TBRC657.

| Molasses Conc. | Total Cell Weight | SOD Activity |
|---|---|---|
| 5 | 1.86 ± 0.04 | 332.57 ± 21.08 [d] |
| 10 | 2.05 ± 0.09 | 329.91 ± 27.66 [d] |
| 15 | 2.35 ± 0.07 | 350.88 ± 30.70 [d] |
| 20 | 2.08 ± 0.06 | 421.10 ± 40.58 [c] |
| 25 | 2.05 ± 0.08 | 590.87 ± 39.36 [a] |
| 30 | 1.63 ± 0.04 | 507.39 ± 27.50 [b] |

Different superscripts within the same column indicate significant differences ($p < 0.05$) and values are presented as mean ± SD ($n = 3$).

Another essential factor which is involved in cell biomass and enzyme production is the nitrogen source. To improve growth and SOD production from *S. cerevisiae* TBRC657, the culturing medium was supplemented with different nitrogen-sources, including organic and inorganic nitrogen sources, to achieve a similar amount of yeast cell and SOD production as obtained from normal YPD, as shown in Table 5. After *S. cerevisiae* TBRC657

was harvested and SOD was extracted inside the cells, yeast extract and soytone gave a similar growth of *S. cerevisiae* TBRC657 when compared to normal YPD, as shown in Table 5 (1.45, 1.30 and 1.28 g for yeast extract, soytone and normal YPD, respectively). According to the results, when applied to yeast extract as a sole nitrogen-source for SOD production, *S. cerevisiae* TBRC657 was able to increase SOD production up to 1.24-fold when compared to normal YPD (262.92 and 211.53 U/mL, respectively). Thus, inorganic nitrogen-sources such as urea or sodium nitrate which were used in this study were not suitable nitrogen-sources for the selected strain due to a very low cell and SOD production of *S. cerevisiae* TBRC657. Thus, yeast extract is the most common component applied as a nitrogen-source in culture media due to a high content of amino acids and essential nutrients for cell growth [34,35]. After varying the concentration of yeast extract ranging from 1.0% (*w/w*)–5.0% (*w/w*), the cell growth and SOD production were constant once 3.0% (*w/w*) yeast extract was applied (data not shown), which would be used for further studies.

**Table 5.** Effect of nitrogen source on SOD production from *S. cerevisiae* TBRC657.

| Nitrogen Source | Total Cell Weight | SOD Activity |
|---|---|---|
| Normal YPD | 1.28 ± 0.04 | 211.53 ± 3.77 [b] |
| Peptone | 0.62 ± 0.01 | 81.08 ± 8.34 [d] |
| Tryptone | 1.06 ± 0.02 | 115.18 ± 9.83 [c] |
| Yeast extract | 1.45 ± 0.03 | 262.96 ± 15.92 [a] |
| Malt extract | 0.61 ± 0.01 | 25.03 ± 1.38 [e] |
| Soytone | 1.30 ± 0.03 | 70.84 ± 0.15 [d] |
| Urea | 0.42 ± 0.01 | 13.47 ± 4.71 [e,f] |
| Sodium nitrate | 0.48 ± 0.02 | 8.80 ± 2.59 [f] |
| Ammonium sulfate | 0.00 | 0.00 |
| Albumin | 0.76 ± 0.01 | 10.50 ± 0.58 [e,f] |

Different superscripts within the same column indicate significant differences ($p < 0.05$) and values are presented as mean ± SD ($n = 3$).

After investigating the medium composition for SOD production, molasses and yeast extract were found to be suitable carbon- and nitrogen sources for SOD production from *S. cerevisiae* TBRC657. The optimal concentration of molasses and yeast extract were 25.0% (*w/w*) and 3% (*w/w*), respectively, which were used as the major composition of the optimized medium in Table 6. To enhance SOD production, some chemical and physical conditions were tested to increase oxidative stress, leading to a higher production of SOD and other antioxidants, which will eliminate the excess free radicals inside the cell. According to previous reports, some chemicals such as hydrogen peroxide, menadione or acetic acid could at the appropriate concentration induce SOD productivity from many microorganisms [36–38]. According to the results obtained, the optimized medium having molasses as major component, which consisted of many types of sugar and chemicals at high concentrations, acted as a hypertonic solution which caused osmotic pressure inside the cells [39]. This condition will generate osmotic and oxidative stresses inside *S. cerevisiae* TBRC657, resulting in an increase in SOD production. Before this experiment, varying the concentration of hydrogen peroxide and menadione, which maximize SOD production without any effect on the growth of the selected strain, was examined. The results exhibited that the suitable concentrations of hydrogen peroxide and menadione were 100 mM and 1 mM, respectively (data not shown). As expected, those inducers at the desired concentrations enhanced SOD production up to 1.37-fold compared to the non-induced control when culturing the selected strain in normal YPD. This result indicates that the chemicals used in this study significantly enhanced SOD production from *S. cerevisiae* TBRC657. On the other hand, applying those inducers to the cell while culturing in the optimized medium did not enhance SOD production and also effected the growth of the yeast cells due to the decreasing value of total cell weight when compared to the non-induced culture, as shown in Table 6. In addition, the optimized medium without any chemical induction increased SOD production from *S. cerevisiae* TBRC657

up to 3.97-fold when compared to the normal YPD. Thus, the SOD production using the optimized medium provided many advantages, such as being easy to operate and not requiring any inducers, which might reduce the overall cost of enzyme production. According to the results obtained, the optimized medium was used for investigating SOD production in a 5 L bioreactor in further experiments.

**Table 6.** Effect of type and concentration of inducer on SOD production from *S. cerevisiae* TBRC657.

| Medium | Inducer | Total Cell Weight | SOD Activity |
|---|---|---|---|
| YPD | - | $1.27 \pm 0.05$ | $211.28 \pm 21.42$ [c] |
| | 100 mM $H_2O_2$ | $1.02 \pm 0.04$ | $274.39 \pm 23.51$ [c] |
| | 1 mM menadione | $0.99 \pm 0.08$ | $289.61 \pm 22.53$ [c] |
| Optimized medium | - | $2.21 \pm 0.03$ | $839.35 \pm 23.70$ [a] |
| | 100 mM $H_2O_2$ | $2.18 \pm 0.04$ | $637.01 \pm 15.64$ [b] |
| | 1 mM menadione | $2.25 \pm 0.01$ | $812.06 \pm 12.63$ [a] |

Different superscripts within the same column indicate significant differences ($p < 0.05$) and values are presented as mean $\pm$ SD ($n = 3$).

### 3.4. Optimization of SOD Production in 5 L Bioreactor

Once the final composition of the optimized medium was successfully established, the aeration rate was investigated to obtain the highest SOD production from *S. cerevisiae* TBRC657. The results exhibited that the aeration rate was a significant factor for the growth of *S. cerevisiae* TBRC657 and SOD production, as shown in Table 7. When the system did not supply any aeration flux, the amounts of yeast cell and SOD activity were the lowest compared to the other conditions. Thus, at an aeration rate of 0.25 vvm, the cell production with the highest SOD activity of 106.04 g and 1235.54 U/mL, respectively, was exhibited. Even though the growth of *S. cerevisiae* TBRC657 at 1.00 vvm gave the highest value (116.11 g), the SOD production was similar when culturing under 0.25 and 0.50 vvm (1078.80, 1117.03 and 1235.54 U/mL for 1.00, 0.50 and 0.25 vvm, respectively). Due to economic feasibility, the lower aeration rate (0.25 vvm) giving the highest SOD production was selected for the production protocol to supply excess oxygen in the system. Moreover, the optimized medium and production process need to be further optimized for SOD production on the pilot-scale in order to reach technical and economic feasibility.

**Table 7.** Effect of aeration rate on SOD production from *S. cerevisiae* TBRC657 in 5 L bioreactor.

| Aeration Rate | Total cell Weight | SOD Activity |
|---|---|---|
| 0.00 | $64.05 \pm 3.40$ | $491.24 \pm 50.76$ [b] |
| 0.25 | $106.04 \pm 12.72$ | $1,235.54 \pm 89.79$ [a] |
| 0.50 | $110.01 \pm 13.20$ | $1,117.03 \pm 63.05$ [a] |
| 1.00 | $116.11 \pm 9.28$ | $1,078.80 \pm 67.22$ [a] |

Different superscripts within the same column indicate significant differences ($p < 0.05$) and values are presented as mean $\pm$ SD ($n = 3$).

### 3.5. SOD Formulation and Storage Stability

Before a thermal shift assay was conducted, the obtained SOD was partial purified by ammonium sulfate precipitation ranging from 40–60% of saturation, which exhibited the highest enzyme yield and purification factor. A DSF study of 4 different groups of additives—polymers and polyols, salts, sugar alcohols and sugars—at various concentrations, involving differences of conformational structure and biological function of the target enzymes, exhibited that each additive at various concentrations had a different effect on the structure of the partial purified SOD. Thus, the various values of ΔTm were revealed in this study. Positive ΔTm represents additives enabling protection of conformational structure of the target enzyme, whereas negative ΔTm represents the additives disrupting the conformational structure of the target enzyme [40]. Among these additives, a mixture between the partial purified SOD and 1.0% (*w/v*) trehalose exhibited the highest ΔTm of

15.17 °C compared to the unformulated SOD, as shown in Table 8. The additives that provided the second and third highest ΔTm were 10.0% (*v/v*) glycerol and 1.0% (*w/v*) ammonium chloride, which exhibited ΔTm values of 14.00 °C and 13.50 °C, respectively, compared to the unformulated SOD. According to the results, most additives used in this study gave positive ΔTm values when compared to the unformulated control, suggesting that those additives were able to protect the obtained SOD and stabilize its conformational structure in a harsh environment. Thus, those additives can enhance the strength of the hydrophobic interactions between non-polar amino acid residues in the protein, leading to a more rigid protein, which promotes enzyme stability in those environments. The different groups of additives which are reported to provide a different function for stabilizing the conformational structure of the target enzyme were examined and mixed with the obtained enzyme. For example, sugars such as fructose, glucose or trehalose can enhance the bonding strength of hydrophobic interaction between the enzyme itself and the others, causing a stable structure of the target enzyme which can protect the enzyme from harsh environments. Salt solutions such as sodium chloride or ammonium chloride can protect the structure of the enzyme via competition in the binding between proteins and water molecules. Thus, water molecules will be removed from the target protein, forming a more compact structure. Some additives can form a hydrophobic interaction with the target enzyme, leading to reduced movement and also reduced water activity of the enzyme, which causes the formulated enzymes to be more stable in the environment [41–44]. Thus, these top 3 formulated SODs were selected and further analyzed for their SOD activities and storage stability at 4 °C for 6 months.

**Table 8.** Additives and their concentration giving the highest difference of melting temperature compared to unformulated SOD.

| Group | Additive | ΔTm |
|---|---|---|
| Polymers and Polyols | 0.1% (*w/v*) ethylene glycol | 0.00 ± 0.00 [d] |
| | 10.0% (*v/v*) glycerol | 14.00 ± 0.10 [a] |
| | 1.0% (*w/v*) PEG1000 | 5.25 ± 0.05 [c] |
| | 10.0% (*w/v*) PEG3350 | 0.33 ± 0.01 [d] |
| | 0.1% (*w/v*) PEG8000 | –0.17 ± 0.01 [d,e] |
| Salt | 10.0% (*w/v*) NaCl | 2.67 ± 0.01 [c] |
| | 10.0% (*w/v*) KCl | 3.67 ± 0.01 [c] |
| | 0.1% (*w/v*) CaCl$_2$ | 0.00 ± 0.00 [d] |
| | 0.1% (*w/v*) MgCl$_2$ | 1.50 ± 0.01 [d] |
| | 1.0% (*w/v*) NH$_4$Cl | 13.50 ± 0.25 [a] |
| | 10.0% (*w/v*) MnCl$_2$ | 9.67 ± 0.01 [b] |
| Sugar alcohol | 0.1% (*w/v*) Mannitol | 0.50 ± 0.01 [d] |
| | 0.1% (*w/v*) Sorbitol | 0.51 ± 0.01 [d] |
| | 10.0% (*w/v*) Xylitol | 4.50 ± 0.01 [c] |
| Sugar | 1.0% (*w/v*) Fructose | 10.67 ± 0.01 [b] |
| | 10.0% (*w/v*) Maltose | 3.67 ± 0.01 [c] |
| | 0.1% (*w/v*) Sucrose | 3.50 ± 0.01 [c] |
| | 1.0% (*w/v*) Trehalose | 15.17 ± 0.50 [a] |
| | 0.1% (*w/v*) Mannose | 0.00 ± 0.00 [d] |

Different superscripts within the same column indicate significant differences ($p < 0.05$) and values are presented as mean ± SD ($n = 3$).

The top 3 formulates providing the highest ΔTm—1.0% (*w/v*) trehalose, 10.0% (*v/v*) glycerol and 1.0% (*w/v*) ammonium chloride—were selected and measured for their SOD activities in order to investigate the effect of those additives on biological function. Thus, the suitable additives should provide a high ΔTm with the same level of SOD activity compared to the unformulated SOD. The results show that all 3 formulates exhibited an SOD activity of 971.42 U/mL, 988.14 U/mL and 1002.67 U/mL, respectively, whilst the unformulated SOD showed the same level of SOD activity of 982.44 U/mL (Table 9). Thus, the results indicated that the selected additives were involved in the conformational structure of

the target enzyme but did not affect the activity of the enzyme. The formulated SODs were stored for 6 months and the SOD activity from each formulate was measured every month compared to the unformulated SOD, which was set as 100.00% since the beginning of experiment. The results from all selected formulates stored for up to 6 months, show that all formulates, especially trehalose or glycerol, seemed to have a potential to stabilize SOD activity at storage conditions, resulting in more than 80% of the remaining SOD activity being detected when compared to the unformulated control, which dramatically decreased in activity since month 3 of the experiment. The remarkable increase in storage stability with the selected additives might be explained by preferential exclusion theory, suggesting that the addition of additives—trehalose or glycerol—to the enzyme solution segregates the water molecules away from the enzyme surface, resulting in a reduction of the hydration radius and increasing the compactness of the enzyme molecules which consequently stabilize the target enzyme. Thus, the enhancement of the conformational structure to enhance storage stability might require more than one group of additives which have different functions on the target enzyme [41,45,46]. Mixing different groups of additives might have a synergistic effect, resulting in enhanced efficiency of storage stability of the target enzyme by altering its conformational structure, which can be detected by other biophysical techniques in further studies.

**Table 9.** The remaining SOD activity obtained from 3 formulates compared to the unformulated version after storing at 4 °C for 6 months.

| Formulation | Remaining SOD Activity | | | | | | |
|---|---|---|---|---|---|---|---|
| | 0 Months | 1 Month | 2 Months | 3 Months | 4 Months | 5 Months | 6 Months |
| Native SOD | 100.00 ± 1.99 [a] | 92.33 ± 2.45 [a,b] | 92.44 ± 5.14 [ab] | 65.46 ± 2.23 [d] | 21.77 ± 1.98 [e] | 0.28 ± 0.01 [f] | 0.00 ± 0.00 [f] |
| 1.0% (*w/v*) trehalose | 98.88 ± 4.48 [a] | 92.53 ± 2.35 [ab] | 94.53 ± 4.28 [a] | 97.05 ± 4.58 [a] | 88.31 ± 3.36 [b] | 89.40 ± 3.78 [b] | 86.49 ± 1.22 [b] |
| 10.0% (*w/v*) glycerol | 100.58 ± 5.47 [a] | 97.93 ± 8.09 [a] | 95.95 ± 3.41 [a] | 95.69 ± 2.57 [a] | 87.05 ± 3.37 [b] | 86.13 ± 2.63 [b] | 85.98 ± 1.34 [b] |
| 1.0% (*w/v*) NH$_4$Cl | 102.06 ± 6.88 [a] | 102.06 ± 6.98 [a] | 91.88 ± 2.90 [ab] | 87.86 ± 5.41 [b] | 82.81 ± 4.48 [b] | 81.28 ± 3.68 [b] | 75.75 ± 2.04 [bc] |

Different superscripts within the same column indicate significant differences ($p < 0.05$) and values are presented as mean ± SD ($n = 3$).

### 3.6. Cytotoxicity Activity and Protective Effect of SOD on Fibroblast Cells

The obtained SOD is expected to be used as a bioactive ingredient in the cosmetic and personal care product industries. For that reason, the cytotoxicity and proliferation of the obtained SOD from *S. cerevisiae* TBRC657 was tested against fibroblast cells, which represent cells that synthesize the extracellular matrix and collagen in connective tissue. After 24 h of treatment, the percentage of living cells treated with an SOD dosage of 50 U and 100 U increased, while dramatically decreasing when treated with higher dosages (Table 10). This result shows high cell viability and fibroblast cell proliferation when treated with a smaller dosage of SOD, with more than 80% of cell viability, which makes it declarable as a non-toxic substance to fibroblast cells. Then, the inhibitory concentration 50% or IC50, which indicates the concentration which is required to inhibit 50% of the proliferation, was examined, and the result showed that IC50 of the obtained SOD was approximately 2660 U, which is equal to 1.77 mg/mL protein of crude extract (data not shown). Thus, the results obtained indicate that a small dosage of the obtained SOD enhanced a proliferation of fibroblast cells. A treatment at 500 units of SOD to fibroblast cells exhibited a cell viability of 98.77% (Table 10), which makes it declarable as non-toxic to fibroblast cells because more than 80% of cell viability was detected [47,48]. According to the results, the cell viability was predicted to be decreased when treating with higher dosages of crude SOD, indicating the limitations of crude SOD dosages.

**Table 10.** Cell viability after treated with various dosages of SOD by MTT assay.

| SOD Dosage | Cell Viability |
|:---:|:---:|
| 0 | $100.00 \pm 3.98$ [a,b] |
| 50 | $114.12 \pm 5.79$ [a] |
| 100 | $108.08 \pm 0.78$ [a,b] |
| 200 | $100.44 \pm 5.64$ [a,b] |
| 500 | $98.77 \pm 2.83$ [ab] |

Different superscripts within the same column indicate significant differences ($p < 0.05$) and values are presented as mean $\pm$ SD ($n = 3$).

Following this, the protective effect of the obtained SOD on fibroblast cells was observed. Firstly, the effect of hydrogen peroxide on cell morphology and viability was studied using SOD-untreated fibroblast cells. The morphology of those fibroblast cells directly treated with the highest concentration of hydrogen peroxide showed shrinking and no attachment to others, which is characteristic of programmed cell death with a viability of 53.50% compared to the control. Hence, adding an exogenous hydrogen peroxide caused the fibroblast cells to generate a high level of ROS inside the cells, followed by a programmed cell death phenomenon [49]. Thus, the protective effect of SOD-treated cells on ROS was investigated. After incubating fibroblast cells with various dosages of SOD overnight, the cells were treated with exogenous hydrogen peroxide for 1 h, with amounts ranging from 12.5–200.0 µg/mL, causing ROS formation inside the fibroblast cells. Then, cell viability of all treatments was measured by MTT assays and compared to the untreated cells. According to Figure 1, a high dosage of SOD (200 units and 500 units) enhanced the protective effect on ROS inside the cell generated by 100 µg/mL of hydrogen peroxide, with an approximate 20% increase in cell viability compared to the other treatments. This result indicates that the obtained SOD had a protective effect on exogenous hydrogen peroxide [50]. On the other hand, 200 µg/mL of hydrogen peroxide was an excess concentration which triggered programmed cell death, even though the highest dosage of SOD was not sufficient [51,52]. This result indicates that SOD obtained from *S. cerevisiae* TBRC657 plays an essential role in oxidative stress protection, with a protective effect on ROS inside the cells which can be used and applied in many industrial applications such as cosmetics and personal care products.

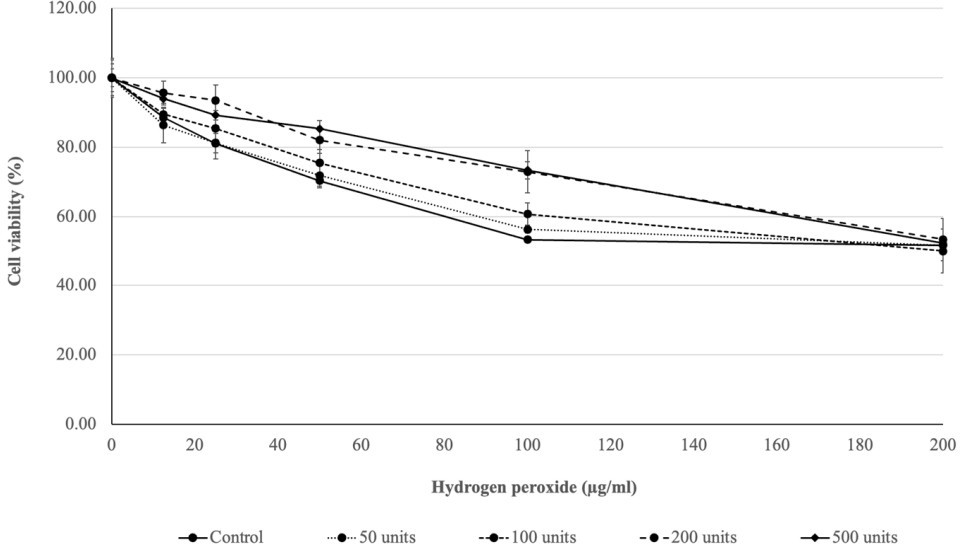

**Figure 1.** Protective effect of SOD on hydrogen peroxide-treated fibroblast cells.

## 4. Conclusions

SOD is an essential antioxidant enzyme which eliminates and balances free radicals inside cells. Due to its function, SOD can be applied in many industries. In this study, *S. cerevisiae* TBRC657 showed the highest SOD production and was selected to investigate the optimization of the SOD production process, especially medium optimization, in order to develop a cost-effective process. The optimized medium, consisting of molasses and yeast extract, exhibited a high production of SOD from the selected strain without any chemical induction, due to the properties and suitable concentration of molasses. The obtained optimized medium provided many advantages such as ease of operation and not requiring any inducers, which might reduce the overall cost of enzyme production, and it can be applied in large-scale production. Thus, mixing the obtained SOD with compatible additives, especially trehalose and glycerol, protected the conformational structure of the enzyme, which resulted in a prolonged storage stability of the obtained SOD under storage conditions. Moreover, SOD at small dosages enhanced the proliferation of fibroblast cells with a protective effect on ROS inside the cells by adding exogenous hydrogen peroxide. Thus, SOD obtained from *S. cerevisiae* TBRC657 produced by culturing in the optimized medium could be a promising candidate for use as a ROS scavenger, which can be applied in many industries.

**Author Contributions:** Conceptualization, P.P., S.S., M.K., T.N. and P.S.; methodology, P.P., K.S., J.A. and A.J.; validation, P.P., T.N. and P.S.; writing—original draft preparation, P.P., K.S., T.N. and P.S.; writing—review and editing, P.P., T.N. and P.S.; funding acquisition, T.N. and P.S. All authors have read and agreed to the published version of the manuscript.

**Funding:** This research received funding by National Center for Genetic Engineering and Biotechnology (BIOTEC) under the research project "Development of Specialty Enzymes for Eco-Industrial Innovative Products" (grant number: P-18-50932) and Biotechnology of Biopolymers and Bioactive Compounds Special Research Unit, Department of Biotechnology, Faculty of Agro-Industry, Kasetsart University.

**Institutional Review Board Statement:** Not applicable.

**Informed Consent Statement:** Not applicable.

**Data Availability Statement:** Not applicable.

**Acknowledgments:** The author would like to thank all staff in the Enzyme Technology Research Team and the Biotechnology of Biopolymers and Bioactive Compounds Special Research Unit for their kindly help and support in numerous discussions.

**Conflicts of Interest:** The authors declare no conflict of interest.

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
