# Peer review of "Enhancing the Productivity and Stability of Superoxide Dismutase from Saccharomyces cerevisiae TBRC657 and Its Application as a Free Radical Scavenger"

_fermentation, doi:10.3390/fermentation8040169_

Round 1

Reviewer 1 Report

The authors show data on the different steps for production, isolation and stabilization of SOD from a specific strain of S. cerevisiae. The manuscript is well written, although it could be strongly reduced without impacting on the reader comprehension of the main message.

major concern:

-  the data on the protective effect of SOD on cell viability of fibroblast, or its putative toxic effect, lacks statistical analysis for a real understanding of the differences observed.  Moreover, the data on the protective effect of SOD on the oxidative induced stress by hydrogen peroxide on fibroblasts is over interpreted. I am not convinced that the authors have data to say that “SOD had the potential to eliminate the ROS produced inside the cell…. “(pag 475 line 464-466). In my opinion this data should be left out of the manuscript as its interpretation is questionable, and it does not add any other layer of complexity in the data presented.

- in the tables it is missing the meaning of the small letters (a, b, ….) probably are related with statistical analysis.
